# Dynamic Changes in Qidan Aroma during Roasting: Characterization of Aroma Compounds and Their Kinetic Fitting

**DOI:** 10.3390/foods13111611

**Published:** 2024-05-22

**Authors:** Ying Wang, Yue Duan, Huanlu Song

**Affiliations:** Laboratory of Molecular Sensory Science, School of Food and Health, Beijing Technology and Business University (BTBU), 11 Fucheng Road, Beijing 100048, China; wangying_96118@163.com (Y.W.); angellfatumai@163.com (Y.D.)

**Keywords:** SPME-GC×GC-O-MS, aroma-active compounds, Maillard products, dynamic changes, kinetic fitting

## Abstract

Qidan is one of the most famous varieties of Wuyi Rock tea and has a strong aroma. The aroma-active compounds in Qidan subject to different roasting times were analyzed using solid-phase microextraction two-dimensional gas chromatography–olfactometry–mass spectrometry (SPME-GC×GC-O-MS), and a total of 92 aroma-active compounds were detected. Multivariate statistical analysis showed that the roasting time had a significant effect on the aroma characteristics of Qidan, and that the key products in the Maillard reaction accumulated with the extension of the roasting time; these key products were screened out according to the calculation of the odor activity values (OAVs), from which kinetic equations were established. It was found that the levels of 2-methylbutanal, 3-methylbutanal, 2-methylpyrazine, 2-ethyl-5-methylpyrazine, and benzaldehyde increased with time, while the contents of benzeneacetaldehyde showed a tendency to first increase and then decrease. This study provides a theoretical basis for flavor quality control during Qidan processing.

## 1. Introduction

Tea is an important beverage in Chinese culture, playing an important role in cultural exchange and economic development. Wuyi Rock tea (WRT), a type of oolong tea, has a unique ‘rock flavor’ due to various factors, such as the variety, climate, and production process [1]. According to the Chinese national standard, Dahongpao is the most representative variety of WRT and is considered to be one of the four famous traditional teas [2]. Qidan has an elegant aroma, with a slightly spicy and mellow sweet taste, making it a treasure of WRT [3].

Aroma is a crucial factor in evaluating the quality of tea. More than 700 volatile compounds have been identified to come from tea, including aldehydes, alcohols, ketones, esters, and heterocyclic compounds [4]. Yue et al. [5] studied the aroma characteristics of 16 oolong tea varieties and identified 368 volatile compounds using the headspace solid-phase microextraction–gas chromatography–mass spectrometry (HS-SPME-GC/MS) technique. The volatile compounds with a floral aroma have the highest r-OAVs, such as *β*-ionone, geraniol, benzene acetaldehyde, benzoic acid methyl ester, and indole.

Roasting is the most crucial factor affecting the aroma of tea, causing the cleavage and isomerization of some insoluble substances, which improves its mellow taste and pure aroma [6]. According to Zhang et al. [7], oolong tea can enhance the flavor of summer tea by controlling the moisture content. This combination of teas has been found to effectively remove the stale taste, improve the aroma, and reduce the bitterness and astringency. Roasting is a key process that is linked to the quality of oolong tea [8]. Appropriate measures must be taken during the drying process to ensure smooth thermochemical, decomposition, and Maillard reactions, resulting in the production of more aromatic substances. During the roasting process, most of the aromatic substances produced during pre-fermentation should be retained as much as possible, while the main aromatic substances that determine the aroma of WRT should be selectively fixed [8].

The Maillard reaction, also known as the carbonyl ammonia reaction [9,10], is the main non-enzymatic browning reaction that occurs during baking in food technology. The Amadori rearrangement products produced during this process can react with ammonia and undergo cyclization to form pyrazines and pyrroles. In addition, they can be directly cyclized to form aromatic compounds, such as furans and their derivatives. The Amadori products can also go on to react with amino acids to produce Strecker degradation products, such as Strecker aldehydes and sulfur-containing compounds [11].

Yang et al. [12] investigated the Maillard reaction products formed during the roasting process of WRT. They identified several indicator compounds, including furfural, 5-methyl-2-furancarboxaldehyde, 1-ethyl-1H-pyrrole-2-carboxaldehyde, 2-ethyl-5-methylpyrazine, 1-ethyl-1H-pyrrole, 2-ethyl-6-methylpyrazine, and methylpyrazine, which can be used to monitor the roasting process. Additionally, it has been suggested that 3,5-dimethyl-2-ethylpyrazine may contribute to the aroma of Shui Xian after roasting [3]. Guo et al. [13] emphasized the importance of 2-methylpyrazine and 2,5-dimethylpyrazine in their study on the aroma of WRT. Yue et al. [14] performed a non-targeted analysis of volatile aromatic compounds (VFCs) and targeted aroma-active compounds in Wuyi Rock tea from four different cultivars using GC-O/MS. They found that methyl salicylate, terephthalimide, 2,5-dimethylpyrazine, and 1-furfurylpyrrole in DHP were found to be significantly related to the “floral and fruity”, “green and refreshing”, “baked and caramelized”, “sweetness” and “herbaceous” attributes.

In summary, the Maillard reaction is important in exploring the basic scientific issues related to the roasting process of WRT. While many reports have investigated the effect of temperature on the aroma of WRT, few have focused on the effect of time. Additionally, most studies have used Dahongpao samples rather than pure-bred Dahongpao Qidan samples. Therefore, in this study, the aims were as follows: (1) to evaluate the aroma profiles of Qidan under different roasting time conditions by using an electronic nose system; (2) to identify the differences in the aroma compounds and their respective characteristics in samples roasted for different lengths of time by using SPME and comprehensive two-dimensional gas chromatography–olfactometry–mass spectrometry (GC×GC-O-MS); and (3) to establish the Maillard reaction kinetic equation for the main aroma compounds. Therefore, this work is expected to reveal the effects of roasting time on the aroma of Qidan and to provide scientific guidance for tea production.

## 2. Materials and Methods

### 2.1. Preparation of Tea Samples

Qidan is produced in the Bishiyan area of the Wuyi Mountains, located in Fujian Province, with the geomagnetic coordinates 117.57° E and 27.42° N. All samples were provided by Qiwei Tea Co., Ltd. (Wuyishan, China) and were roasted according to GB/T 18745-2006 [15], with the roasting temperature controlled at 110 ± 5 °C. The samples (500 g) were taken from three roasting cages every 2 h. The final samples of Qidan were obtained at 0 h, 2 h, 4 h, 6 h, 8 h, 10 h, and 12 h. The samples were sealed in an aluminum foil bag and placed in a refrigerator at 4 °C for subsequent analysis.

### 2.2. Chemicals and Reagents

A variety of chemicals and reagents were employed in this study, sourced from multiple suppliers as detailed as follows below:

1. From Sigma-Aldrich (Shanghai, China): 2-methylbutanal, 3-methylbutanal, furfural, 2-ethyl-3-methylpyrazine, 2-methyl-3-heptanone, and n-alkanes (C_7_-C_30_);

2. From TCL (Shanghai, China): 2-methylpyrazine, 2-ethylpyrazine, 1-ethyl-1H-pyrrole-2-carboxaldehyde, benzaldehyde, benzeneacetaldehyde, and methyl pyrrol-2-yl ketone;

3. From J&K Scientific (Shanghai, China): 2-ethyl-3,6-dimethylpyrazine;

4. From MREDA (Beijing, China): 2-Methyl-5-formylfuran;

5. From TM Standard (Changzhou, China): 2-ethyl-6-methylpyrazine and pyrrole-2-carboxaldehyde;

6. From Yuanye Bio-Technology (Shanghai, China): 2-ethyl-5-methylpyrazine and 2-vinylpyrazine;

7. From ACMEC (Shanghai, China): 2-ethyl-3,5-dimethylpyrazine, 2-methyl-3,5-diethylpyrazine, and 1-Furfurylpyrrole;

8. From Jiuding Chemical (Shanghai, China): 1-furfurylpyrrole.

The additional reagents include NaCl (99.5% purity) from Sinopharm Chemical Reagent Co., Ltd. (Shanghai, China); hexane (98% purity) from Fisher Chemicals (Shanghai, China); and gases such as nitrogen (99.999%), liquid nitrogen, and ultra-high-purity helium (99.9992%) from Beijing Asia-Pacific Baifu Gas Industry Co. (Beijing, China).

### 2.3. E-Nose Analysis

The volatile compounds in different varieties of Qidan were analyzed using a portable electronic nose system, PEN3 (Airsense Analytics GmbH). The test time was set to 200 s per sample. The ten aroma receptors of the e-nose are W1C, W1W, W1S, W2W, W2S, W3C, W3S, W5S, W5C, and W6S, which are sensitive to functional groups “aromatic hydrocarbons”, “broad ranges (in particular nitrogen oxides)”, “aromatic ammonia”, “hydrogen”, “aromatic fats”, “broad methane”, “sulfuric organics”, and “broad alcohols”, as well as sensitive functional groups “sulfur chlorine” and “methane fats”, respectively [1]. Each sample was analyzed ten times.

### 2.4. Solid-Phase Microextraction (SPME)

Prior to the formal analysis, tea infusions were prepared for each tea sample, in accordance with the Chinese national standard “Sensory Evaluation Methods for Tea” (GB/T 23776-2018) [16]. The procedure involved weighing the tea samples (2 g), brewing them with boiling water (100 mL) (tea/water ratio: 1:50), and covering with a lid for 3 min, before weighing the tea broth samples (5 g), placing them into a 30 mL headspace vial, and adding NaCl (1.5 g) and 2-methyl-3-heptanone (1 μL) (0.816 μg/μL, dissolved in n-hexane). The vials were incubated at 55 °C for 20 min in a thermostatic water bath (Shanghai Ge Trading Co., Ltd., Shanghai, China). Then, SPME extraction was performed using 2 cm of divinylbenzene/carboxen/polydimethylsiloxane (DVB/CAR/PDMS) extraction fiber (50/30 µm, Supelco, Bellefonte, PA, USA) at 55 °C for 40 min [17], before the extraction fiber was inserted into the inlet of the gas chromatography instrument and desorbed at 230 °C for 5 min.

### 2.5. GC×GC–O–MS Analysis

The GC×GC-O-MS instrumentation utilized in this study included an Agilent 8890 GC-5977B MS, equipped with a primary polar DB-WAX capillary column (30 m × 0.25 mm, 0.25 µm film thickness; Agilent Technologies, Bejing, China) and a secondary medium-polarity DB-17 MS column (1.85 m × 0.18 mm, 0.18 µm film thickness; Agilent Technologies). The method used was based on the one described by Yang [17], with appropriate modifications. The temperature was initially set at 40 °C and held for 3 min, before being increased to 230 °C at a rate of 4 °C/min and held for 5 min. The inlet temperature was 230 °C and helium was used as the carrier gas, with a flow rate of 1 mL/min. The mass spectrometry conditions were as follows: the ion source temperature was 230 °C, the transmission line temperature was 280 °C, and the temperature of the quadrupole was 150 °C. The electron bombardment ionization mode was employed, with a scanning range of m/z 29~500 and an electron energy of 70 eV. Additionally, a sniffing port (Sniffer 9000; Brechbühler, Schlieren, Switzerland) was installed between the GC and MS to transport aroma-active compounds to the human nose using an ultra-high-purity helium carrier gas at a flow rate of 1 mL/min. The olfactometry part was composed of three trained people, who described the aroma and recorded the time that it flowed out. Each sample was analyzed three times.

### 2.6. Aroma Extraction Dilution Analysis (AEDA)

The dilution of the aroma-active compounds was achieved by controlling the volume of gas purged into the sniffing device, through adjusting the split ratio of the carrier gas during the purge process (splitless or 1:1, 3:1, 7:1, or 15:1). The diluted extracts were then analyzed under the same experimental conditions until the maximum dilution at which a particular odor could be smelled was reached. The maximum dilution multiple was the dilution factor of the compound corresponding to that odor, i.e., the flavor dilution (FD) factor. Each sample was tested in three parallel experiments. 

### 2.7. Odor Activity Value (OAV)

The OAV is calculated from the ratio of the concentration of the aroma-active compound to the threshold value. The threshold values for different aroma-active compounds are referenced in a previous study [18]. In general, aroma-active compounds with an OAV greater than or equal to 1 are considered to contribute to the overall odor of the sample.

### 2.8. Qualitative and Quantitative Analysis

The aroma-active compounds (FD ≥ 8) were quantified using gas chromatography–triple quadrupole mass spectrometry (GC-QqQ, Agilent 7890 A GC-7000 MS) in selective ion mode (SIM), according to the results of AEDA. The external standard method with internal standard correction [17] was employed. To reduce errors, the standard compounds were prepared and diluted with n-hexane. The mixed standard solution was configured based on the difference in the retention time or peak area between the compounds, to ensure that each solution contained a similar level of compounds for the GC. Nine appropriate concentration gradients were used, and 1 μL of 2-methyl-3-heptanone (0.816 μg/μL) was added to each as an internal standard compound. The standard curve was drawn according to the ratio of the GC peak area of both the standard compound and the internal standard substance to the added concentration. Then, the peak area ratio of each compound in the sample to the internal standard substance was determined and, with the concentration of the internal standard substance, was used to calculate the compound concentration [17]. To ensure the accuracy of the results, the analysis of both the standard compound solution and the sample was taken as the average of three results.

### 2.9. Kinetic Studies Analysis

With the increase in the osmosis time, the concentrations of the main aroma-active compounds in tea changed by different degrees. To investigate the formation pattern of these characteristic compounds, their kinetics were further investigated with reference to previous experimental methods [19,20]. The kinetic formulae are as follows:(1)$$\frac{dA}{dt}=K\times A^{n}$$

The letters A, t, K, and n in the formula represent the content of volatile compounds, the reaction time, the reaction rate constant, and the reaction energy level, respectively. By integrating kinetic Equation (1), we obtained the following linear Equation (2):(2)$$A=A_{0}+K_{t}$$

A_0_ represents the initial concentration.

The first-order kinetic equation is as follows:(3)$$\ln A=\ln A_{0}+K_{t}$$

The fit of the regression curve between the measured values and model calculations is represented by R^2^, where the closer the value of R^2^ is to 1, the closer the model is to the measured values. 

### 2.10. Statistical Analysis

All the results are expressed as the mean ± standard deviation (average ± SD). The mean and ANOVA of all the aromatic compounds were collated and analyzed using Excel 2019 and SPSS Statistics 27 software. Statistical significance was determined at the *p* ≤ 0.05 level. Figures were created using OriginPro 2023 b software. Partial least square discriminant analysis (PLS-DA) and principal component analysis (PCA) were performed using the SIMCA-P14.1 software.

## 3. Results and Discussion

### 3.1. Results of E-Nose Analysis

Figure 1A displays a radargram that was derived from the e-nose data. The seven samples exhibit similar spindle-shaped profiles, but the ten types of metal oxide sensors differ in intensity. The seven samples responded most strongly to the W5S receptor, which is very sensitive to nitrogen oxides, suggesting that this sensor is the most crucial indicator for distinguishing between these samples. Moreover, nitrogen oxides are linked to Maillard-related products. After 12 h of roasting, the response value of the W5S sensor for Qidan was significantly higher than that of the other samples. This may be due to the accumulation of nitrogen oxides produced by the Maillard reaction, as the roasting time is increased [17]. In Figure 1B, the X and Y axes represent the contribution rates of PC1 (99.03%) and PC2 (0.57%), respectively. The larger the contribution rate, the better the PC performance and the better it reflects the original information. Therefore, it is evident that samples roasted for different lengths of time can be distinguished from one another, with slight but noticeable variations between the 0 h, 2 h, and 4 h samples, as well as smaller yet distinguishable differences between the 8 h, 10 h, and 12 h samples. In conclusion, the electronic nose results indicate that different roasting times produce samples with distinct flavor profiles. The variation in odor may be attributed to the nitrogen oxides generated by the Maillard reaction.

### 3.2. Analysis of Aroma-Active Compounds in Qidan

In this study, 92 aroma-active compounds were detected in seven samples that were analyzed using SPME-GC×GC-O-MS. Table 1 shows the semi-quantitative concentrations of each sample. The compounds were classified into one of the following seven groups: aldehydes (26), alcohols (20), heterocycles (15), esters (12), ketones (9), acids (6), and alkenes (4). Table 1 shows that with an increase in the roasting time, the number of heterocyclic compounds gradually increases, while the number of other compound types remains relatively constant. During the manufacturing process, the Maillard reaction produces a significant amount of heterocyclic compounds, such as furans, pyrroles, thiophenes, and their derivatives, in addition to Strecker degradation products, sulfur-containing compounds, and so on [17]. According to Guo et al. [3], 3,5-dimethyl-2-ethylpyrazine may contribute to the aroma of Narcissus after roasting. The difference in flavor between Dahongpao and Jieding Oolong after roasting has been attributed to the higher content of both 2,5-dimethylpyrazine and trimethylpyrazine in Dahongpao [21]. Additionally, Yuan et al. [22] found that 2-methylpyrazine is an important metabolite for Shuixian, allowing it to present a woody fragrance. 

The type and content of aldehydes also affect the aroma of tea. Peter Schieberle’s team [23] used molecular sensory science techniques to identify 2/3-methylbutyraldehyde as the key aroma-active compound in Darjeeling black tea. The above studies have confirmed the important contribution of Maillard reaction products to the characteristic aroma of WRT. However, the key aroma-active compounds related to the Maillard reaction in Qidan are unknown, and the effects of different roasting times on these compounds have rarely been reported.

### 3.3. Analysis of Maillard Reaction-Related Products in Qidan

A total of 20 key aroma-active compounds (FD ≥ 8) related to the Maillard reaction were screened in the Qidan samples and were quantified (Table 2).

As shown in Table 2, the amount and concentration of aroma-active compounds related to the Maillard reaction increased with the increase in roasting time, with the lowest number being at 0 h, but increasing at 12 h. Moreover, 2-Ethyl-6-methylpyrazine, 2-ethyl-3-methylpyrazine, and ethenylpyrazine started to be produced after 8 h of roasting; 2-ethylpyrazine and 2-ethyl-3,6-dimethylpyrazine were identified after 10 h; whereas 2-ethyl-3,5-dimethylpyrazine was only identified at 12 h. It was demonstrated that pyrazine production increases significantly at 70 °C and reaches its maximum rate at 120 °C. However, when heated at 100 °C, it takes 4 h to reach its maximum content [24]. These findings indicate that the duration of the roasting process can impact the production of pyrazines; as the roasting time is increased at a certain temperature, the number of pyrazine species will increase, and the content will accumulate.

In order to further investigate the changes in the flavor-active compounds of Qidan during roasting, multivariate statistical analyses were performed based on the concentrations of 20 key aroma-active compounds in the Maillard reaction (Figure 2). The results of the principal component analysis (PCA) are shown in Figure 2A. Samples with different roasting times are categorized into the following three categories: the first category includes 0 h, 2 h, 4 h, and 6 h (green); the second category includes 8 and 10 h (blue); and the third category includes 12 h (red). This indicates that there are differences in the aroma characteristics of Qidan at different roasting times. Figure 2B is a biplot with different elliptical areas indicating the 95% confidence intervals. As the roasting time increases, the concentrations of most aroma-active compounds reach their maximum value, and the VIP value indicates their relevance. Figure 2C details the aromatic compounds with VIP values greater than 1, including 2-ethyl-6-methylpyrazine, 2-ethyl-3,5-dimethylpyrazine, furfural, 2-ethylpyrazine, benzeneacetaldehyde, 2-ethyl-3,6-dimethylpyrazine, and ethenylpyrazine. These substances are the main reason for the differences in the aroma of Qidan at different roasting times. These compounds have all been reported in studies on aroma-active substances after the roasting of WRT [7,25]. The replacement test in Figure 2D shows that the model was successfully validated and that both Q^2^ and R^2^ were higher than 0.5. The samples in these three categories had significant differences that could be differentiated from each other, which indicated that the roasting process had a more significant effect on the aroma composition and release of Qidan, which is in line with the findings of Wang et al. and Zhan et al. [2,26].

### 3.4. Modeling the Generation of Characteristic Differential Aroma Compounds

Kinetic equations were developed to investigate the release pattern of key aroma-active compounds resulting from the Maillard reaction at different roasting times. Ge [19] selected glucose and lysine to establish kinetic models to simulate the Maillard reaction in the processing system of brown sugar, in order to study the effects of the processing time and temperature conditions on the main flavor-active compounds, such as aldehydes, pyrazines, and furans, as well as to provide a theoretical basis for their actual production through the establishment of kinetic equations. Yang [17] explored the generation of aroma-active substances in each system under medium-fire roasting conditions by modeling glucose and theanine, as well as glucose and glutamic acid. The glutamic acid system produced more pyrazines, including 2,5-dimethylpyrazine, 5-methyl-2-ethylpyrazine, 2,3,5-trimethylpyrazine, and 2,5-dimethyl-3-ethylpyrazine, compared to the theanine system. Most previous studies have simplified the model of the Maillard reaction by selecting key sugars and amino acids in order to establish it. However, the Maillard reaction is an extremely complex process, and the simplified model does not simulate the change rule of the precursor substances and intermediates in the real products. Therefore, in this paper, we selected Qidan as the research object to study its key aroma-active compounds. The most important aroma-active compounds in the Maillard reaction process were screened out through the OAV calculation (Table 3), resulting in six aroma-active compounds with average OAV values of ≥1, which were 2-methylbutanal, 3-methylbutanal, 2-methylpyrazine, 2-ethyl-5-methylpyrazine, benzaldehyde, and benzeneacetaldehyde.

The production and generation patterns of these compounds were fitted, as shown in Table 4. In Figure 3, the concentrations of the key differential aroma-active compounds changed by varying degrees in all simulated systems, as the roasting time was extended. Moreover, 2-Methylbutanal, 3-methylbutanal, 2-methylpyrazine, 2-ethyl-5-methylpyrazinee, and benzaldehyde exhibited a similar generation pattern during the roasting process, with their concentrations increasing as the roasting time was extended, reaching their maximum values after 12 h. The correlation coefficients obtained by fitting the first-order kinetic equations were significantly higher, with R^2^ values close to 1, based on the concentration of the compounds as a function of time. These key aroma-active compounds are closely related to the Maillard reaction and present a roasted aroma. Guo et al. [13] found that 5-methyl-2-ethylpyrazine is the key aroma-active compound in WRT. In addition, 2-Methylbutanal and 3-methylbutanal are also essential compounds in the aroma profile of WRT [23]. The contribution of benzaldehyde to the aroma of WRT has also been emphasized in the research by Wang et al. [2] as well. Benzeneacetaldehyde is an important contributor to the floral aroma of Qidan, and its content showed a tendency to increase and then decrease during the roasting process, reaching the maximum at 10 h. The formation of benzeneacetaldehyde is also closely related to lipid oxidation; Zamora [27] found a similar pattern of change in their study investigating the effect of the roasting time on the aroma of cinnamon, concluding that benzeneacetaldehyde is an important phenylalanine volatile compound in tea leaves. When phenylalanine and lipid oxidation products were heated together, benzaldehyde and benzeneacetaldehyde were produced, the amount of which depended on the lipid oxidation products involved.

The study above shows that the Maillard reaction’s key aroma-active compounds accumulate as the roasting time extends, except for benzeneacetaldehyde, which reaches its maximum amount of content at 10 h. To ensure the harmony of Qidan’s aroma, controlling the roasting time during production is crucial. It is important to note that this study is limited by its sample size. Therefore, further studies are needed to expand the sample size and verify the effect of roasting time on the flavor of Qidan samples. Additionally, a kinetic model should be established to validate the above conclusions.

## 4. Conclusions

In conclusion, this study identified the characteristics of aroma-active compounds during the roasting process of Qidan through molecular sensory analysis. Differences in the odor of samples with different roasting times were identified using an electronic nose. The samples were analyzed by clustering through PCA and OPLS-DA. The OAVs were calculated to identify the key aroma-active compounds in the Maillard reaction, with kinetic equations being developed for the six key compounds. The results showed that the roasting process had a more positive effect on the aroma-active components of Qidan, resulting in a richer overall flavor of the samples. The concentrations of 2-methylbutyraldehyde, 3-methylbutyraldehyde, 2-methylpyrazine, 5-methyl-2-ethylpyrazine, and benzaldehyde increased with the increase in roasting time. In contrast, the concentration of phenylacetaldehyde showed a trend of first increasing and then decreasing, reaching its maximum concentration after 10 h. Therefore, it is particularly important to control the roasting time in the production process, to ensure the richness of the aroma. In order to ensure the harmony of the Qidan aroma, the roasting time should not exceed 10 h. An in-depth study of Qidan samples with different roasting times was carried out to eliminate empiricism in the traditional tea roasting process and to provide a theoretical basis for this purpose.

## Figures and Tables

**Figure 1 foods-13-01611-f001:**
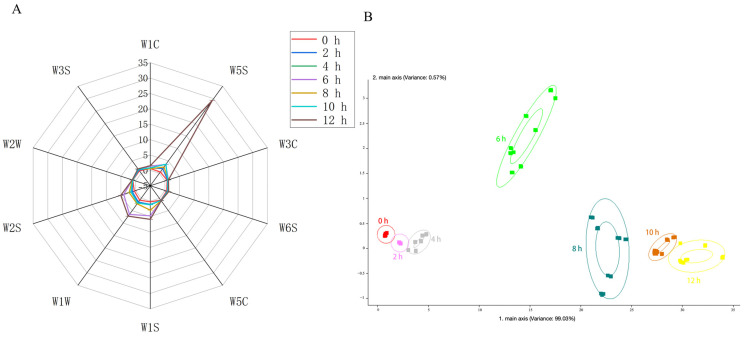
Radargrams and PCA analyses of the e-nose. (**A**) Radargrams of e-nose responses for different types of volatile compounds at different roasting times; (**B**) e-nose score plots (PCA) at different roasting times.

**Figure 2 foods-13-01611-f002:**
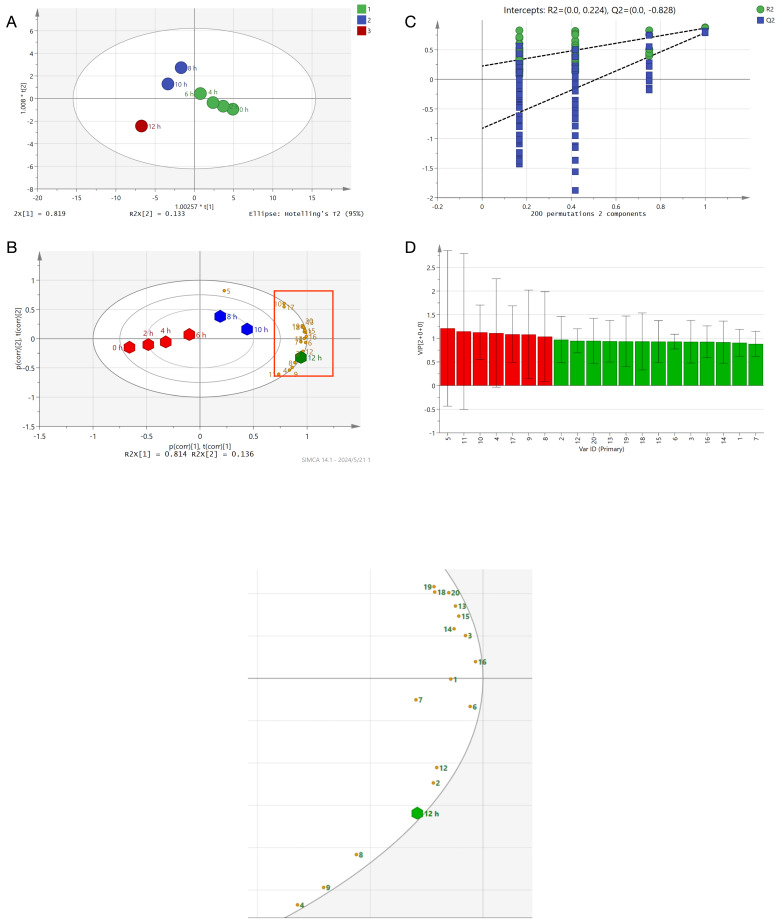
OPLS-DA plots based on SPME-GC×GC-MS data results: (**A**) is a scatter plot of the predicted scores, the green, blue, and red signals represent the Qidan samples with different roasting times, respectively; (**B**) is a biplot, with different elliptical areas indicating 95% confidence intervals; (**C**) represents the variable importance in projection (VIP); and (**D**) represents the validated model.

**Figure 3 foods-13-01611-f003:**
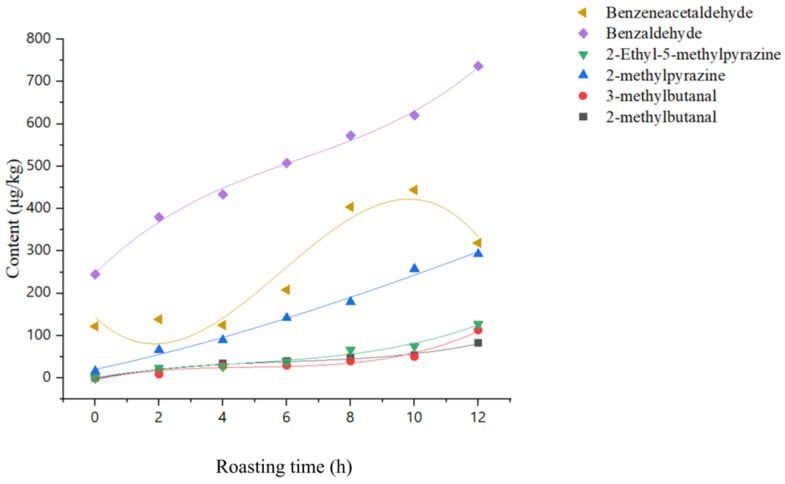
Variation pattern of the concentration of key aroma-active compounds with time.

**Table 1 foods-13-01611-t001:** The aroma-active compounds and their FD factors detected in Qidan using SPME-GC×GC-O-MS.

CAS	Compound	Perception ^1^	RI ^2^	Concentration (μg/kg)							Identification Method ^3^
				0 h	2 h	4 h	6 h	8 h	10 h	12 h	
Aldehyde											
96-17-3	2-Methylbutanal	cocoa, almond	901	-	8	8	32	32	64	128	MS/RI/O/STD
590-86-3	3-Methylbutanal	cocoa	910	-	8	16	16	32	32	128	MS/RI/O/STD
110-62-3	Pentanal	cocoa	935	16	16	16	8	8	16	8	MS/RI/O
66-25-1	Hexanal	grass, fatty	1083	32	32	16	16	8	8	8	MS/RI/O
1576-87-0	(*E*)-2-pentenal	fruity	1131	32	16	16	16	8	2	2	MS/RI/O
111-71-7	Heptanal	fatty	1181	64	64	32	32	16	16	8	MS/RI/O
6728-26-3	(*E*)-2-hexenal	fruity	1201	8	8	4	4	4	4	4	MS/RI/O
505-57-7	2-Hexenal	fruity	1216	8	8	8	16	16	16	8	MS/RI/O
6728-31-0	4-Heptenal	creamy	1236	32	32	16	16	16	8	8	MS/RI/O
18829-55-5	(*E*)-2-heptenal	fatty	1243	8	8	4	4	4	2	2	MS/RI/O
124-13-0	Octanal	fatty	1280	8	8	8	4	4	4	4	MS/RI/O
124-19-6	Nonanal	fatty	1385	16	16	16	8	8	8	4	MS/RI/O
142-83-6	(*E*,*E*)-2,4-Hexadienal	sweet	1397	32	32	32	16	16	8	8	MS/RI/O
2548-87-0	(*E*)-2-octenal	fatty	1434	8	8	16	16	16	8	4	MS/RI/O
98-01-1	Furfural	bread	1455	16	64	64	128	128	64	64	MS/RI/O/STD
4313-03-5	(*E*,*E*)-2,4-heptadienal	fatty	1493	32	32	32	32	32	32	32	MS/RI/O
112-31-2	Decanal	orange	1500	4	4	4	4	8	4	4	MS/RI/O
100-52-7	Benzaldehyde	almond	1534	64	64	64	64	64	128	128	MS/RI/O/STD
620-02-0	5-Methylfurfural	caramel	1560	-	32	64	64	64	128	128	MS/RI/O/STD
18829-56-6	(*E*)-2-nonenal	soapy, cucumber	1582	2	2	2	2	2	2	2	MS/RI/O
432-25-7	*β*-Cyclocitral	herbal	1590	16	8	8	16	16	16	32	MS/RI/O
2167-14-8	1-Ethyl-1H-pyrrole-2-carboxaldehyde	burnt	1616	-	8	16	16	32	64	64	MS/RI/O/STD
122-78-1	Phenylethanal	honey, sweet	1625	32	64	64	64	64	64	32	MS/RI/O/STD
25152-84-5	(*E*,*E*)-2,4-decadienal	fatty	1819	16	16	8	8	4	4	2	MS/RI/O
2363-88-4	2,4-Decadienal	orange	1824	8	8	8	16	8	16	8	MS/RI/O
4411-89-6	*α*-Ethylidenbenzeneacetaldehyde	floral	1907	16	8	8	4	8	16	16	MS/RI/O
Alcoholic											
616-25-1	1-Penten-3-ol	pungent	1158	8	8	8	16	16	16	8	MS/RI/O
71-41-0	1-Pentanol	fusel	1255	16	8	8	8	8	4	2	MS/RI/O
1576-95-0	(*Z*)-2-penten-1-ol	green, spicy	1310	16	8	8	8	4	4	4	MS/RI/O
111-27-3	1-Hexanol	resin, flower	1363	32	32	32	32	64	32	8	MS/RI/O
544-12-7	3-Hexen-1-ol	green leafy	1384	4	4	2	2	2	2	2	MS/RI/O
928-95-0	(*E*)-2-hexenol	green	1392	16	8	8	4	4	2	2	MS/RI/O
3391-86-4	1-Octen-3-ol	mushroom	1394	32	16	16	8	8	4	2	MS/RI/O
5989-33-3	(*Z*)-linalool oxide	flower	1420	32	8	8	8	4	4	2	MS/RI/O
111-70-6	Heptanol	green	1447	32	32	16	16	16	8	4	MS/RI/O
104-76-7	2-Ethyl-1-hexanol	fatty	1487	16	16	16	16	16	8	4	MS/RI/O
78-70-6	Linalool	flower	1537	32	32	16	16	8	8	4	MS/RI/O
111-87-5	1-Octanol	chemical	1545	16	16	16	16	16	8	8	MS/RI/O
29957-43-5	Dehydrolinalool	moldy	1648	32	32	32	16	16	16	32	MS/RI/O
98-00-0	Furfuryl alcohol	burnt	1659	64	64	64	128	128	256	256	MS/RI/O
39028-58-5	(*E*)-linalool oxide (pyranoid)	woody	1741	32	32	16	16	8	8	2	MS/RI/O
106-25-2	Nerol	sweet	1767	256	256	256	128	128	64	64	MS/RI/O
106-24-1	Geraniol	floral	1860	128	128	128	64	64	32	8	MS/RI/O
100-51-6	Benzyl alcohol	sweet, flower	1877	128	128	128	128	64	64	128	MS/RI/O
60-12-8	Phenethyl alcohol	rosy	1912	128	64	64	64	32	32	32	MS/RI/O
40716-66-3	*α*-Nerolidol	floral	2017	128	128	64	64	32	32	16	MS/RI/O
Ketone											
1629-58-9	1-Penten-3-one	pungent	973	32	32	16	16	8	16	8	MS/RI/O
2408-37-9	2,2,6-Trimethyl-cyclohexanone	honey	1335	2	2	2	2	2	2	2	MS/RI/O
1669-44-9	3-Octen-2-one	creamy	1414	16	16	8	8	8	8	4	MS/RI/O
14309-57-0	3-Nonen-2-one	spicy	1506	8	8	4	4	4	2	2	MS/RI/O
30086-02-3	(*E*,*E*)-3,5-octadien-2-one	fruity	1590	8	8	4	2	2	2	2	MS/RI/O
127-41-3	*α*-Ionone	floral	1863	32	16	16	16	8	8	2	MS/RI/O
79-77-6	*β*-Lonone	floral	1917	64	16	16	16	8	8	2	MS/RI/O
488-10-8	(*Z*)-jasmone	floral	1972	64	16	16	16	16	8	2	MS/RI/O
14901-07-6	*β*-Ionone	floral	1977	16	8	16	8	8	8	8	MS/RI/O
Esters											
13894-62-7	Methyl (*Z*)-3-hexenoate	fruity	1253	-	2	2	2	2	2	2	MS/RI/O
35154-45-1	(*Z*)-3-Hexen-1-yl isovalerate	fruity	1440	32	16	16	16	16	16	8	MS/RI/O
6378-65-0	Hexyl Hexanoate	fruity	1599	16	8	8	8	4	4	2	MS/RI/O
31501-11-8	3-Hexenyl hexanoate	fatty	1646	64	64	32	32	32	32	16	MS/RI/O
53398-86-0	(*E*)-2-hexen-1-yl hexanoate	fruity	1660	16	16	16	16	8	8	4	MS/RI/O
695-06-7	*γ*-Caprolactone	creamy	1694	16	16	16	8	8	4	4	MS/RI/O
61444-38-0	(*Z*)-(*Z*)-Hex-3-en-1-yl hex-3-enoate	fruity	1715	8	8	4	4	2	2	2	MS/RI/O
119-36-8	Methyl salicylate	peppermint	1745	64	64	32	32	16	8	2	MS/RI/O
101-41-7	Methyl 2-phenylacetate	floral	1749	8	8	8	4	16	16	8	MS/RI/O
103-45-7	Phenylethyl acetate	floral	1829	8	8	16	16	8	8	8	MS/RI/O
103-48-0	*β*-Phenylethyl isobutyrate	fruity	1877	8	8	8	8	8	8	4	MS/RI/O
103-52-6	2-Phenylethyl butanoate	fruity	1958	8	8	4	4	4	4	2	MS/RI/O
Acid											
109-52-4	Pentanoic acid	sweat	1090	4	2	2	2	2	2	2	MS/RI/O
142-62-1	Hexanoic acid	sour fatty	1829	8	8	4	4	4	2	2	MS/RI/O
112-05-0	Nonanoic acid	cheesy, fat	2174	16	16	8	8	4	4	2	MS/RI/O
25524-95-2	(*Z*)-7-Decen-5-olide	fatty	2273	4	4	4	4	2	2	2	MS/RI/O
334-48-5	Decanoic acid	rancid, fat	2316	8	8	4	4	2	2	2	MS/RI/O
79-09-4	Propanoic acid	acidic	1525	16	8	-	-	-	-	-	MS/RI/O
Alkene											
123-35-3	*β*-Myrcene	woody, citrus	1170	2	2	2	2	2	2	2	MS/RI/O
5989-27-5	(*+*)-Limonene	citrus	1201	8	8	8	4	4	2	2	MS/RI/O
28973-97-9	*β*-(*Z*)-Farnesene	citrus	1660	8	8	4	4	4	4	4	MS/RI/O
502-61-4	*α*-Farnesene	herbal	1748	8	4	4	4	2	2	2	MS/RI/O
Heterocyclic											
534-22-5	2-Methylfuran	chocolate	876	-	2	4	8	8	16	16	MS/RI/O
109-08-0	2-Methylpyrazine	nutty	1176	16	32	63	128	128	128	256	MS/RI/O/STD
3777-69-3	2-Pentylfuran	fruity	1250	2	2	4	4	8	8	8	MS/RI/O
13925-00-3	2-Ethylpyrazine	nutty	1333	-	-	-	-	-	8	64	MS/RI/O/STD
13925-03-6	2-Ethyl-6-methylpyrazine	roasted hazelnut	1385	-	-	-	-	16	8	-	MS/RI/O/STD
13360-64-0	2-Ethyl-5-methylpyrazine	nutty	1392	-	16	16	16	64	64	128	MS/RI/O/STD
15707-23-0	2-Ethyl-3-methylpyrazine	nutty	1414	-	-	-	-	16	32	32	MS/RI/O/STD
4177-16-6	Ethenylpyrazine	burnt nutty	1434	-	-	-	-	8	16	32	MS/RI/O/STD
13360-65-1	2-Ethyl-3,6-dimethylpyrazine	hazelnut	1435	-	-	-	-	-	8	16	MS/RI/O/STD
13925-07-0	2-Ethyl-3,5-dimethylpyrazine	coffee	1464	-	-	-	-	-	-	64	MS/RI/O/STD
1192-62-7	2-Furyl methyl	nutty	1490	-	-	-	8	8	32	64	MS/RI/O/STD
18138-05-1	2-Methyl-3,5-diethylpyrazine	nutty	1497	-	8	16	16	32	32	64	MS/RI/O/STD
1438-94-4	1-Furfurylpyrrole	metallic	1833	-	16	16	32	32	64	64	MS/RI/O/STD
1072-83-9	Methyl pyrrol-2-yl ketone	fruity	1971	8	8	8	16	32	32	32	MS/RI/O/STD
120-72-9	Indole	perfumy	2448	64	64	32	32	16	16	4	MS/RI/O

“-”: not detected. ^1^ Aroma perception of each aroma-active compound that was detected at the sniffing port. ^2^ Retention index (RI) on DB-WAX polar column. ^3^ MS, mass spectrum; RI, retention index; O, olfactory; STD, standard compounds. Identification methods with no STD, considered as ‘tentatively identified’.

**Table 2 foods-13-01611-t002:** Quantitative results of 20 aroma-active compounds related to Maillard reaction products at different roasting times.

No.	Compounds	Standard Curves	R^2^	Quota Selected Ions (*m/z*) *	Concentration (μg/kg)						
					0 h	2 h	4 h	6 h	8 h	10 h	12 h
1	2-Methylbutanal	y = 0.3645x + 0.0164	0.997	44, 58, **86**	-	16.96 ± 1.13 ^e^	34.20 ± 4.51 ^d^	39.95 ± 3.31 ^d^	47.39 ± 4.04 ^c^	53.34 ± 5.14 ^b^	82.84 ± 7.66 ^a^
2	3-Methylbutanal	y = 0.3645x + 0.0164	0.997	44, 58, **86**	-	9.18 ± 0.46 ^e^	28.48 ± 3.83 ^d^	30.06 ± 4.58 ^d^	40.14 ± 1.60 ^c^	50.7 ± 2.48 ^b^	113.62 ± 14.73 ^a^
3	2-Methylpyrazine	y = 0.79991x + 0.7851	0.9721	53, 67, **94**	16.53 ± 2.42 ^f^	66.99 ± 8.74 ^e^	89.88 ± 9.76 ^e^	142.29 ± 5.47 ^d^	180.05 ± 13.56 ^c^	258.08 ± 17.05 ^b^	293.05 ± 8.42 ^a^
4	2-Ethylpyrazine	y = 0.1241x − 0.1042	0.9952	53, 80, **107**	-	-	-	-	-	36.25 ± 4.18 ^b^	137.98 ± 14.47 ^a^
5	2-Ethyl-6-methylpyrazine	y = 0.3296x + 0.0012	0.9973	56, 94, **121**	-	-	-	-	55.67 ± 7.79 ^a^	12.81 ± 1.91 ^b^	-
6	2-Ethyl-5-methylpyrazine	y = 0.3296x + 0.0012	0.9964	56, 94, **121**	-	24.6 ± 2.22 ^e^	27.46 ± 1.31 ^e^	39.29 ± 5.67 ^d^	66.58 ± 8.62 ^c^	75.01 ± 7.57 ^b^	127.42 ± 11.86 ^a^
7	2-Ethyl-3-methylpyrazine	y = 0.4531x + 0.1372	0.9924	67, 93, **121**	-	-	-	-	27.9 ± 2.28 ^b^	56.24 ± 4.16 ^a^	68.62 ± 5.93 ^a^
8	Ethenylpyrazine	y = 0.3032x + 0.0309	0.9932	52, 79, **106**	-	-	-	-	10.68 ± 1.81 ^c^	21.94 ± 3.39 ^b^	71.64 ± 5.09 ^a^
9	2-Ethyl-3,6-dimethylpyrazine	y = 0.5021x + 0.0924	0.9917	81, 108, **135**	-	-	-	-	-	12.98 ± 1.37 ^b^	35.3 ± 6.17 ^a^
10	Furfural	y = 0.3999x + 0.0182	0.9947	67, **96**	84.61 ± 8.87 ^f^	118.14 ± 17.43 ^ef^	207.59 ± 16.52 ^d^	411.76 ± 61.81 ^c^	683.08 ± 28.66 ^a^	553.27 ± 43.85 ^b^	492.95 ± 76.03 ^c^
11	2-Ethyl-3,5-dimethylpyrazine	y = 0.4171x + 0.0201	0.985	80, 108, **135**	-	-	-	-	-	-	162.88 ± 16.66
12	2-Furyl methyl	y = 0.2485x − 0.0005	0.991	43, 67, **95**	-	-	-	11.64 ± 0.75 ^d^	26.4 ± 2.32 ^c^	60.68 ± 5.94 ^b^	100.64 ± 13.53 ^a^
13	2-Methyl-3,5-diethylpyrazine	y = 0.4196x + 0.0009	0.9913	67, 122, **149**	-	5.31 ± 0.37 ^f^	33.11 ± 2.53 ^e^	50.68 ± 2.72 ^d^	69.66 ± 4.53 ^c^	81.33 ± 7.32 ^b^	101.71 ± 12.29 ^a^
14	Benzaldehyde	y = 0.223x + 0.0674	0.9964	51, 77, **106**	244.92 ± 14.68 ^d^	379.99 ± 11.65 ^c^	433.91 ± 19.77 ^b^	507.83 ± 44.48 ^b^	572.92 ± 56.56 ^b^	620.59 ± 80.64 ^a^	736.85 ± 72.86 ^a^
15	2-Methyl-5-formylfuran	y = 1.0196x + 0.0019	0.9997	53, 81, **110**	-	76.91 ± 12.9 ^de^	135.25 ± 23.12 ^d^	305.05 ± 44.45 ^c^	356.02 ± 25.44 ^b^	458.28 ± 36.72 ^b^	555.78 ± 63.86 ^a^
16	1-Ethyl-1H-pyrrole-2-carboxaldehyde	y = 1.8519x + 0.0059	0.9967	66, 94, **123**	-	18.99 ± 1.99 ^f^	36.76 ± 4.75 ^e^	60.56 ± 10.39 ^d^	77.45 ± 13.67 ^c^	116.01 ± 32.92 ^b^	145.6 ± 16.24 ^a^
17	Benzeneacetaldehyde	y = 0.9824x + 0.0112	0.9941	65, 91, **120**	122.28 ± 17.84 ^d^	138.94 ± 22.29 ^d^	125.17 ± 17.09 ^d^	208.35 ± 23.54 ^c^	404.2 ± 12.15 ^a^	444.44 ± 18.94 ^a^	319.07 ± 35.06 ^b^
18	1-Furfurylpyrrole	y = 1.2413x − 0.0521	0.9973	53, 81, **141**	-	24.02 ± 1.05 ^f^	41.19 ± 2.57 ^e^	60.9 ± 4.23 ^d^	71.14 ± 6.92 ^c^	94.07 ± 13.73 ^b^	101.05 ± 24.28 ^a^
19	Methyl pyrrol-2-yl ketone	y = 1.19171x + 0.0022	0.9966	66, 94, **109**	1.69 ± 0.13 ^f^	13.44 ± 0.98 ^e^	24.2 ± 1.74 ^d^	35.24 ± 0.18 ^c^	42.45 ± 2.09 ^b^	47.03 ± 2.13 ^b^	56.2 ± 5.16 ^a^
20	Pyrrole-2-carboxaldehyde	y = 0.4578x − 0.0984	0.9932	39, 66, **95**	-	0.36 ± 0.05 ^e^	5.67 ± 0.42 ^d^	11.97 ± 1.47 ^c^	16.34 ± 2.19 ^b^	20.27 ± 2.53 ^a^	23.48 ± 1.31 ^a^

* Ions for SIM mode quantitative analysis are selected as coarsened to represent the mother ion with the largest molecular weight. Values in the same line followed by different superscript letters are significantly different (*p* < 0.05).

**Table 3 foods-13-01611-t003:** OAVs of 20 aroma-active compounds related to Maillard reaction products at different roasting times.

No.	Compounds	Perception	Odor Threshold in Water (μg/kg)	OAV						
				0 h	2 h	4 h	6 h	8 h	10 h	12 h
1	2-Methylbutanal	cocoa, coffee	0.35	-	48	97	114	135	152	236
2	3-Methylbutanal	cocoa	0.35	-	26	81	85	114	144	324
3	2-Methylpyrazine	nutty, popcorn	60	<1	1	1	2	3	4	4
4	2-Ethylpyrazine	nutty	4000	-	-	-	-	-	<1	<1
5	2-Ethyl-6-methylpyrazine	nutty	40	-	-	-	-	1	<1	-
6	2-Ethyl-5-methylpyrazine	coffee, nutty	16	-	1	1	2	4	4	7
7	2-Ethyl-3-methylpyrazine	roasted	130	-	-	-	-	<1	<1	<1
8	Ethenylpyrazine	roasted nutty	700	-	-	-	-	<1	<1	<1
9	2-Ethyl-3,6-dimethylpyrazine	roasted	0.4	-	-	-	-	-	32	88
10	Furfural	toast	770	<1	<1	<1	<1	<1	<1	<1
11	2-Ethyl-3,5-dimethylpyrazine	roasted, nutty	0.04	-	-	-	-	-	-	4072
12	2-Furyl methyl	cocoa, coffee	10,000	-	-	-	<1	<1	<1	<1
13	2-Methyl-3,5-diethylpyrazine	nutty	1040	-	<1	<1	<1	<1	<1	<1
14	Benzaldehyde	bitter almond	24	10	15	18	21	23	25	30
15	2-Methyl-5-formylfuran	caramel	500	-	<1	<1	<1	<1	<1	1
16	1-Ethyl-1H-pyrrole-2-carboxaldehyde	burnt	-	-	-	-	-	-	-	-
17	Benzeneacetaldehyde	honey, floral	0.2	611	694	625	1041	2021	2222	1595
18	1-Furfurylpyrrole	coffee	100	-	<1	<1	<1	<1	<1	1
19	Methyl pyrrol-2-yl ketone	moldy, nutty	58,585	<1	<1	<1	<1	<1	<1	<1
20	Pyrrole-2-carboxaldehyde	beef, coffee	65,000	-	<1	<1	<1	<1	<1	<1

**Table 4 foods-13-01611-t004:** Fitted curves of key aroma-active compounds with different roasting times.

No.	CAS	Compounds	Fitted Equation	R^2^
1	96-17-3	2-Methylbutanal	y = −1.568 + 14.870x − 2.051x^2^ + 0.116x^3^	0.989
2	590-86-3	3-Methylbutanal	y = −3.453 + 15.924x − 3.057x^2^ + 0.210x^3^	0.973
3	109-08-0	2-Methylpyrazine	y = 20.909 + 15.898x + 0.799x^2^ − 0.016x^3^	0.991
4	13360-64-0	2-Ethyl-5-methylpyrazine	y = 0.990 + 12.785x − 1.779x^2^ + 0.132x^3^	0.981
5	100-52-7	Benzaldehyde	y = 247.772 + 73.734x − 7.445x^2^ + 0.39x^3^	0.996
6	122-78-1	Benzeneacetaldehyde	y = 145.001 − 73.599x + 23.595x^2^ − 1.346x^3^	0.925

## Data Availability

The original contributions presented in this study are included in the article, further inquiries can be directed to the corresponding author.

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
