# Peer review of "Dynamic Changes in Qidan Aroma during Roasting: Characterization of Aroma Compounds and Their Kinetic Fitting"

_foods, 2024, doi:10.3390/foods13111611_

Round 1

Reviewer 1 Report (Previous Reviewer 2)

Comments and Suggestions for Authors

Line 28: is Qidan the same as the Dahongpao tea? The same line 235.

Line 36: The proper abbreviation is HS-SPME-GC/MS. Rather than "technology", I would rather say "technique"

Line 37: what means “r-OAVs”?

Line 46-47: re-written this sentence "It is also important to retain the majority of aromatic substances produced during pre-fermentation as much as possible to retain, and selectively fix the main aromatic substances that determine its aroma, should be of considerable value".

Line 51: instead of "industry", I advise you to write "technology".

Line 92: "Chemicals and reagents" should be written better. 

Line 127: the temperature is missing.

Line 346: it is a positive or negative, I mean the accumulation of Maillard reaction products.

"2.5. GC×GC–O–MS Analysis" How many people did the olfactometry part?

In conclusion, there is a lack of recommendations, on when the fermentation should be stopped to achieve the best aroma score.

As it is a food product that is consumed by people, this should be linked with sensory analysis. Moreover, the quality of writing is not satisfied. The manuscript required in general an extensive English editing.

Comments on the Quality of English Language

required English corrections

Reviewer 2 Report (Previous Reviewer 3)

Comments and Suggestions for Authors

Authors submitted a thoroughly revised version of their manuscript by taking into account and by conformed to all-three reviewers' comments and suggestions.

Therefore, I can recommend it, of the manuscript under consideration, in the Journal Foods.

This manuscript is a resubmission of an earlier submission. The following is a list of the peer review reports and author responses from that submission.

Round 1

Reviewer 1 Report

Comments and Suggestions for Authors

The research described in this manuscript gives results that supports the effects of the roasting temperatures evaluated on the aroma of the tea.

In methods it is suggested to describe with more detail the procedures used.

The manuscript is understandable and well written.

The results supports the conclusions stated.

Reviewer 2 Report

Comments and Suggestions for Authors

"Study on the changes and dynamics of aroma compounds in Wuyi Rock Tea (Dahongpao) during the roasting process". It is an interesting topic and a nice methodology, however, I found a description as a lower-quality manuscript

First of all, the abstract uploaded in the system is not the same as prepared in the text. Moreover, any kinds of tables and graphs are included. As it is written it should contain 5 tables and 4 figures.

Second, keywords: “Maillard reaction related products” sounds very long, Authors could work to shorten it, on the other hand also so more keywords could be added related e.g. to the roasting process. Also, the title could sound like this e.g. “Study changes and dynamics of aroma compounds in Dahongpao, variety of Wuyi Rock tea during the roasting process”.

Line 29: The whole name of DHP should be used before the abbreviation is used.

Line 36: what means “r-OAVs”?

Line 31 and 38: The language should be once more reviewed and possibly re-write: “Aroma is a crucial factor…”, then “Roasting is the most crucial factor…”, and Line 43 is a repetition of the first sentence: “Roasting is the key process…”. I suggest working to improve the readiness of the introduction part.

Line 49-60: I see this information as very general and too long/unnecessary due to the topic.

I advise to have a look at these articles: 10.1039/d1fo00165e; 10.3390/foods11244109; 10.1111/1541-4337.12999. They could be helpful to write a good introduction.

Line 125: I could be written as: “2 cm divinylbenzene/…. fiber”

In subchapter 2.5: this sentence has to be rewritten: compounds by carrier gas to the human nose for sniffing using carrier gas”.

Line 206: baking or roasting?

Comments on the Quality of English Language

The manuscript requires English corrections.

Reviewer 3 Report

Comments and Suggestions for Authors

It seems more likely that authors submitted a premature manuscript. Therefore, authors should revise their manuscript appropriately and resubmitt it for consideration, according to the following comments and suggestions:

1) Although authors refer in their Abstract that "The multivariate statistical analysis showed ...", I found nowhere any multivariate statistical analysis, which should have been carried out by using appropriate factorial experimental design. The subsection "2.10. Statistical Analysis" cannot be considered at all as "multivariate statistical analysis". Therefore, authors, in the revised version of their manuscript, should definitely provide an appropriate factorial experimental design.

2)  Although a lot of Figures are reffered within the text, I founf not Figures in the manuscript under consideration, nor any other related file containing the reffered figures; the most of the results are based on them.